# Bycatch survival of shortfin mako sharks (*Isurus oxyrinchus*) in the U.S. Atlantic pelagic longline fishery

Mischa Schultz[1¤], Eric R. Hoffmayer[2], James A. Sulikowski[3], Michael E. Byrne[1]*

1 School of Natural Resources, University of Missouri, Columbia, Missouri, United States of America,
2 Southeast Fisheries Science Center, National Marine Fisheries Service, Pascagoula, Mississippi, United States of America, 3 Coastal Oregon Marine Experiment Station, Oregon State University, Newport, Oregon, United States of America

¤ Current Address: Department of Oceanography and Coastal Sciences, Louisiana State University, Baton Rouge, Louisiana, United States of America

* byrneme@missouri.edu

## Abstract

Severe population declines of shortfin mako sharks (*Isurus oxyrinchus*) in the Atlantic Ocean have led to the implementation of conservation measures, notably fishing retention bans and live-release regulations, aimed at substantially reducing fishing mortality to allow stock recovery. While retention bans can eliminate harvest mortality, their effectiveness can be reduced if survival of sharks encountered as bycatch and not retained is low. We quantified at-vessel survival (AVS) and post-release survival (PRS) and estimated overall bycatch survival probability of mako sharks for the U.S. Atlantic pelagic longline fishery. Based on fisheries observer records ($n = 7821$) between 2000–2020, we found AVS varied regionally from 0.77 (95% CI: 0.74–0.80) in the northmost observation region to 0.65 (95% CI: 0.61–0.69) in the Gulf of Mexico (GOM). We found significant negative correlations between AVS and soak time, surface temperature, mainline length, and shark size. Based on pop-up archival satellite tags ($n = 27$) deployed from pelagic longline vessels in the WNA during 2022–2024, PRS was 0.87 (95% CI: 0.74–0.93). Overall mean bycatch survival probability varied regionally from 0.64 (95% CI: 0.51–0.68) in the northmost observation region to 0.59 (95% CI: 0.49–0.64) in the GOM, which given the low productivity rates of mako sharks may be low enough to hinder recovery efforts if mako sharks are encountered as bycatch in significant numbers. Pairing retention bans with actions that reduce incidence of bycatch would likely provide the greatest benefit to population recovery. Our research highlights the importance of quantifying survival regionally and between fleets, as variability in fishing practices and environmental conditions can result in different bycatch survival outcomes, which can be important considerations in stock assessment.

**Data availability statement:** Tagging information and depth and temperature data from all pop-up archival tags that transmitted data are within the Supporting Information files. Fisheries observer data is not made publicly available to protect private and personally identifiable information. Data queries can be made to the National Marine Fisheries Service's Pelagic Observer Program (https://www.fisheries.noaa.gov/southeast/fisheries-observers/southeast-pelagic-observer-program).

**Funding:** MEB - Quantifying and reducing post-release mortality of shortfin mako sharks (*Isurus oxyrinchus*) captured as bycatch in pelagic long-line fisheries. Grant number: NA19NMF4720227 Funding agency: National Oceanographic and Atmospheric Administration (NOAA). Funding agency website: https://www.noaa.gov/. The funder for this study had no role in study design, data collection and analysis, decision to publish, or preparation of the manuscript.

**Competing interests:** The authors have declared that no competing interests exist.

## Introduction

Overfishing is a primary driver of global shark population declines [1,2]. Even when not targeted directly, bycatch mortality can significantly contribute to overfishing, especially for pelagic shark species whose ranges and habitat use overlap that of other fishes targeted in pelagic longline (PLL) fisheries [3]. The shortfin mako shark, *Isurus oxyrinchus* (hereafter: mako), is a large pelagic shark that is highly vulnerable to overfishing because of its k-selected life history traits such as low productivity and slow growth rate [4]. Unsustainable levels of fishing mortality have previously been reported for mako sharks in the North Atlantic Ocean, where they are common bycatch in commercial PLL fisheries targeting tunas and swordfish [*Xiphias gladius*;[5,6]. Even when caught incidentally, mako sharks have traditionally been retained due to the commercial value of their meat [7]. Consequently, mako sharks in the North Atlantic Ocean have experienced considerable population declines. The most recent North Atlantic stock assessment conducted by the International Committee for the Conservation of Atlantic Tunas (ICCAT) suggested that the stock is overfished, and that overfishing is occurring with high probability [8]. Additionally, the International Union for the Conservation of Nature (IUCN) assessment estimated a~60% decline in mako shark biomass over 75 years [9].

In November 2021, ICCAT adopted management recommendation 21−09, prohibiting the retention of mako sharks captured by contracting parties [10]. In the United States (U.S.), the National Marine Fisheries Service (NMFS) fully implemented the no retention policy for U.S. fishers in the Atlantic Ocean in July 2022 [11]. Retention bans aim to reduce fishing mortality to allow for population recovery by removing direct mortality associated with retaining individuals, and potentially by altering fisher behavior to avoid capturing the species. The success of retention bans as a conservation strategy hinges on a significant proportion of individuals captured surviving fisheries interactions and not contributing to fishing mortality (*F*), which includes all mortalities directly linked to the fishing process [12]. While effectively eliminating mortality from landings, assuming fishers comply with regulations [13,14], retention bans cannot easily mitigate at-vessel mortality (AVM) and post-release mortality (PRM), which cumulatively determine bycatch survival, which is the survival rates of sharks that are hooked but not landed. At-vessel mortality (also referred to as haul back or hooking mortality) represents mortality that occurs before an individual is brought back to the vessel. Post-release mortality (PRM) occurs when a fish released alive later dies because of injuries or physiological stress incurred during capture. At-vessel survival (AVS = 1 − AVM) of mako sharks has been estimated at 0.738 in the Canadian pelagic longline fishery [15], 0.644 in the Portuguese longline fishery [16] and 0.714 in the U.S. Atlantic longline fishery [17]. Based on satellite tagging data, estimates of mako shark post-release survival (PRS = 1 − PRM) rates in the Atlantic Ocean consistently average ~70% [15,18–20], although the U.S. PLL fleet is poorly represented in these studies. Low bycatch survival may undercut the benefits of retention bans for k-selected species such as mako sharks for which relatively low levels of fishing mortality can significantly negatively affect population recovery [4,5].

The probability of a shark surviving the capture and release processes may be influenced by fishing practices (e.g., soak time), environmental factors (e.g., temperature), biological factors (e.g., fish size), handling practices and injuries [16,21–26]. Interactions with mako sharks in the U.S. PLL fleet occur across a wide swath of the western North Atlantic Ocean (Fig 1), encompassing environmental conditions ranging from the warm surface temperatures and deep thermoclines with minimal seasonal variability characteristic of the Gulf of Mexico (GOM) to the spatially and seasonally heterogeneous North Atlantic where the Gulf Stream's warm waters are juxtaposed alongside the continental shelf and the cold waters and the Labrador Current [27]. Fishing practices such as fishing depth and soak time are likely to vary regionally as fishers target different species and adapt to local environmental conditions [28,29]. As regional differences in environmental conditions and fishing practices have potential to result in different levels of mako shark bycatch survival, it is unlikely that survival is homogeneous across the entire U.S. Atlantic PLL fishery.

The objectives of this study were to 1) quantify AVS and PRS of mako sharks in the U.S. Atlantic PLL fishery, 2) determine the factors that influence AVS, and 3) provide regional estimates of overall bycatch survival within the U.S. PLL fishery. Quantifying mako shark bycatch survival can inform future stock assessments and aid in assessing the potential effectiveness of retention bans to facilitate population recovery. Quantifying the factors influencing AVS and the relative contributions of AVS and PRS to bycatch survival may identify where improvements can be made to increase survival

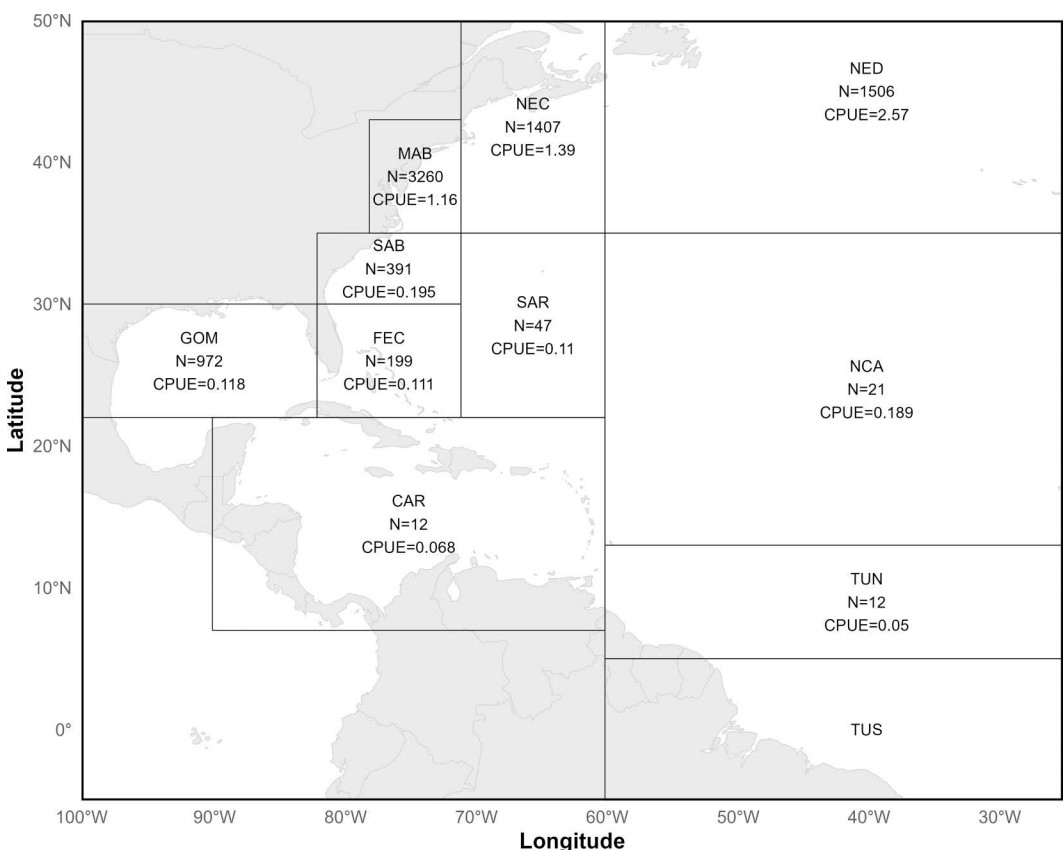

**Fig 1. Pelagic Observer Program mako shark observations.** Pelagic Observer Program statistical regions in the **U.**S. Atlantic defined by the National Marine Fisheries Service. Within each region the total number (N) of shortfin mako shark (*Isurus oxyrinchus*) observations during 2000–2020 are reported along with catch per unit effort (CPUE = N/total sets observed). Statistical regions are Tuna South (TUS), Tuna North (TUN), Caribbean (CAR), North Central Atlantic (NCA), Gulf of Mexico (GOM), Florida East Coast (FEC), Sargasso (SAR), South Atlantic Bight (SAB), Mid Atlantic Bight (MAB), Northeast Coastal (NEC), and Northeast District (NED). World map data from Natural Earth (http://www.naturalearthdata.com/).

rates of mako sharks captured in PLL fisheries [30,31]. Additionally, our study provides the first estimate of mako shark PRS specific to the modern U.S. Atlantic PLL fleet.

## Methods

### At-vessel survival

Mako shark catch data from 2000–2020 were obtained from the U.S. Pelagic Observer Program (POP) administered by the NMFS Southeast Fisheries Science Center (SEFSC). The POP was created in 1992 to monitor the harvest by the U.S. PLL fleet and aid in evaluating pelagic fish stocks. Observers on selected longline vessels record location, target species, environmental data, gear configurations, and information on individual species caught, including size, sex, and whether the animal was alive at haul back. Observer coverage was approximately 4–5% of the total number of reported longline sets by the U.S. PLL fleet prior to 2007 with 8% target coverage annually thereafter [32] and annual reports (available at https://www.fisheries.noaa.gov/national/fisheries-observers/observer-program-reports-and-policies) indicating actual coverage ranged from 10–16% from 2007–2019. There are 11 geographical regions identified by the NMFS for monitoring U.S. PLL fisheries (Fig 1). The majority of POP mako shark observations (98.8%) occurred in six regions: the Gulf of Mexico (GOM, 12.4%), Florida East Coast (FEC, 2.5%), South Atlantic Bight (SAB, 5%), Mid Atlantic Bight (MAB, 41.7%), Northeast Coastal (NEC, 18.0%), and Northeast Distant (NED, 19.2%), with catch per unit effort in these regions (CPUE = mako sharks captured/total sets observed) ranging from 2.57 in the NED to 0.11 in the FEC (Fig 1).

For each mako shark observation, we extracted the recorded state of the shark at haul back as dead (0) or alive (1), along with covariates recorded by observers that may influence AVS. We considered sharks recorded as damaged (i.e., injured from predation before haul back) as dead. Covariates considered included soak time (hours) of the entire line, encompassing the duration of time between when the gear was set until the gear was retrieved. Soak time defines the maximum amount of time a hooked shark may struggle and consequently incur physiological costs and physical injury, and a negative correlation between soak time and AVS has been observed in other pelagic shark species [17,22,23,25]. Leader length (m) was the full length of the gangion from the mainline to the hook. Mako sharks are obligate ram ventilators with large oxygen consumption demands as a function of their high metabolic rates [33,34]. Short leaders may inhibit swimming, limiting ventilation capacity and potentially decreasing survival probability. Additionally, we considered total mainline length (km). Retrieval time increases as mainline length increases, as does the potential distance a hooked shark may be dragged through the water, and thus there may be a negative correlation between mainline length and survival. We included estimated maximum hook depth (m; hereafter: hook depth), as an approximate measure of the depth the gear was fished. This metric is estimated by observers from the combined lengths of leaders and droplines between buoys. We considered shark size (fork length, FL, cm), as susceptibility to physical injury and physiological stress may vary as a function of size and previous studies have reported greater AVS of larger mako sharks [16,17,35]. Sea surface temperature (SST; ºC) was recorded onboard the vessel and was considered because warm temperatures can increase physiological stress responses in sharks [36–38], and as regional endotherms [39] it is plausible mako sharks may be sensitive to overheating if struggling for extended periods in warm waters.

To test whether AVS varied between the 6 regions with > 100 mako shark observations we used mixed effects logistic regression fit using the "lme4" [40] package in R [41] to model AVS with region as a categorical fixed effect (reference category = GOM). We included year as a random effect in all models to account for annual variability and potential bias in observer coverage, and changes in fishing practices and regulations overtime not directly captured in the available data. For example, we did not have data on hook types used, but regulations were put in place requiring use of circle hooks in the U.S. PLL fishery in 2004. We attempted to include set ID as a random effect to account for non-independence of sharks captured on the same set, however given the large proportion of sets in which only a single mako shark was observed (53%), we were not able to achieve model convergence. We could not include vessel ID as a random effect as we did not have access to that information because it confidential and protected. To evaluate the impact of covariates associated with gear, fishing practices

and environmental conditions on mako shark AVS, we used mixed effects logistic regression. As sex was not determined for 22.5% of observed sharks, before fitting models incorporating other covariates we tested for evidence of sex-biased survival by fitting a mixed effects logistic regression model using all observations where sex was identified and including sex as a categorical fixed effect (reference category = female). After ensuring no significant effect of sex on AVS based on the 95% confidence interval (CI) for the parameter estimate (β) of sex crossing 0 (see results), we included all observations in which data on all covariates of interest were recorded by observers, including records where sex was not determined, in further models. We developed a set of 35 candidate models incorporating different combinations of covariates and their interactions, including a null (intercept-only) model. Prior to model construction we calculated correlation coefficients among all covariates and ensured no multicollinearity based on a correlation threshold of | r | > 0.70 [42]. We scaled and centered all covariates to aid model convergence and allow for direct comparisons of covariate effect sizes.

We calculated Akaike's information criterion (AIC) for each model and evaluated relative model support by ranking models by ΔAIC (difference between AIC of model $i$ and the model with the lowest AIC value) and the AIC weight ($w_i$) of each model [43]. We considered models with ΔAIC ≤ 2 as well supported [43]. Following the guidance of Arnold [44], we identified informative parameters in candidate models ranked by AIC as those who's estimated 85% CIs did not cross 0. In the case of model uncertainty (multiple models with ΔAIC ≤ 2), we made inferences based on the model with the lowest ΔAIC in which all parameters were informative. We evaluated performance of the top selected model using leave-one-out cross-validation (LOOCV) in which each observation was sequentially removed, and the model fit with the remaining data was used to predict the survival of the removed shark. We calculated the receiver operating characteristic area under the curve (AUC) statistic [45] which measures overall model predictive accuracy and represents the probability that a model will correctly identify the positive case when presented with a randomly chosen pair of cases in which one is positive and one is negative [46]. An AUC of 1.0 represents perfect prediction, an AUC of 0.50 represents prediction equal to random chance, and an AUC of −1.0 indicates complete failure.

## Post-release survival

Mako sharks captured during normal fishing operations by U.S. PLL vessels were tagged prior to release with pop-up archival tags (PAT; model PSATLIFE or PSATFLEX, Lotek Wireless Inc., Newmarket ON, Canada) either by trained POP observers or cooperating captains during 2022–2024. Observers were instructed to tag all live fish regardless of the likelihood of mortality. Tags were attached via tether to a stainless-steel dart inserted in the dorsal musculature and were applied via a tagging pole while the shark was in the water. Tags were programmed to record temperature (0.2°C accuracy) and depth (±1% full scale accuracy) at 5-minute intervals for 28 days, at which point the tag would detach from the shark, float to the surface, and transmit data through the Argos satellite system. Tags would detach and begin transmission prior to 28 days if the depth remained static for > 72 hours, as would be expected if the tag was floating on the surface following pre-mature detachment or if an animal was dead on the seafloor. These tag models did not include a built-in mechanism to release from the animal if the tag descended below the crush depth (2000 m), and absent of any modification, tags attached to sharks that died beyond the continental shelf and sunk in deep waters would likely be destroyed by water pressure at depth, resulting in no data transmission and potential underestimation of mortalities. Therefore, we added pressure-sensitive emergency release devices (RD1800, Wildlife Computers Inc., Redmond, WA, USA) designed to cut the tether and allow the tag to float to the surface after descending to depths > 1800 m. For each tagged shark, the location, FL, SST, soak time, estimated hook depth, and number of hooks deployed were recorded, as well as the hooking location and estimated length of monofilament leader (estimated to the nearest inch by observers) remaining on release. The shark's condition at haulback was recorded as lethargic if the shark was inactive or moving slowly and active if the shark was swimming and fighting. Release condition was recorded as weak if the shark sank or swam away lethargically or as strong if the shark swam away quickly. Any injuries were noted. Tagging was approved under the University of Missouri Animal Care and Use Protocol 38981.

We examined depth-temperature time-series to determine the fates of tagged sharks. We identified mortalities when a shark sank swiftly and consistently to a depth that triggered the emergency depth release mechanism prior to the end of the programmed 28-day deployment period. Following previous studies [24,28], tags ingested by endothermic animals, as evidenced by variation in depth but maintenance of constant temperature, were classified as mortalities. This conservative approach assumes tag consumption represents a predation event resulting in mortality although it is plausible the shark survived the incident. Tags that floated to the surface prior to 28 days and were not associated with a rapid sinking event or ingestion were considered pre-mature detachments, and sharks that consistently moved throughout the water column for 28 days were considered to have survived. We used a logistic known-fate survival model to estimate daily survival probability. The model took the form:

$$\varnothing_{it} = \frac{e^{(\beta^0)}}{e^{(\beta^0)} + 1}$$

$$\mu_{it} = \varnothing_{it} \times y_{i(t-1)}$$

$$y_{it} \sim Bern(\mu_{it})$$

where $\varnothing_{it}$ is the probability of shark $i$ surviving from day $t$ to day $t+1$, $\beta^0$ is an intercept term, and $y_{i(t-1)}$ is the state of shark $i$ (alive or dead) in day $t-1$. A low incidence of mortalities prevented us from fitting models that included covariates. However, to test for a potential relationship between haul back condition and survival we used Fisher's exact test including data from sharks that died or were tracked for the full 28-day deployment period. We created daily encounter histories for each shark, where for each day the shark was known to have survived (1) or died (0), and sharks were censored from analysis following mortality or pre-mature tag detachment. We fit the model in a Bayesian context using Markov chain Monte Carlo in JAGS [47] via the "runjags" package [48] in R [41]. We specified an uninformative uniform prior for $\beta_0$ and ran 3 MCMC chains of 8000 iterations each, with a burn-in of 6000 iterations and thinning interval of 10. We assessed mixing and model convergence via visual inspection of trace plots and checking that the Gelman-Rubin statistic (R-hat) was < 1.1 [49]. To estimate PRS to 28 days post-release, we exponentiated $\varnothing$ to the 28th power (PRS = $\varnothing^{28}$).

## Bycatch survival

We estimated mako shark bycatch survival, defined as the probability that a hooked mako shark survives both the landing process and lives to 28 days post-release, in each of the 6 regions encompassing the majority of mako shark observations (GOM, FEC, SAB, MAB, NEC, and NED) using Monte Carlo simulation that incorporated results from the AVS and PRS analyses. The steps of which were as follows:

1. We generated a sample population of 1000 hooked sharks, where for each shark ($i$) a random value for each covariate associated with AVS (shark size, hook depth, leader length, soak time, mainline length, and SST) was generated from the observed means and covariance matrix of parameters associated with POP observations of mako shark captures for a region using the mvrnorm function in R [41]. To reduce the probability of improbable values, we set lower bounds on some parameters based on minimum observed values such that, hook depth was > 1.8 m, soak time was > 0.5 hr, and SST was > 6°C, and all sharks were > 70 cm FL based on estimated size at birth [50]. The AVS probability was then predicted for each shark ($i$) from the most supported model of AVS.

2. A PRS probability was randomly pulled from the posterior distribution of the fitted known-fate survival model. A probability of bycatch survival (BS) was then estimated for each shark ($i$) as:

$$BS_i = AVS_i \times PRS$$

Where $AVS_i$ is the probability of shark $i$ being alive at haul-back, and $PRS$ is the probability of a released shark surviving 28 days.

3. We determined the fate (live or die) of each simulated shark by a random draw from a binomial distribution with a probability equal to $BS_i$, and calculated population-level bycatch survival as the proportion of the 1000 simulated sharks to survive. We ran this simulation procedure 1000 times for each region. The Monte Carlo simulation approach was advantageous as it facilitated propagation of model uncertainty while providing regional distributions of bycatch survival estimates based on observed fishing and environmental conditions characteristic of each region.

## Results

### At-vessel survival

Observers recorded data from 17394 longline sets from 2000–2020 and observed 7821 mako shark captures (Fig 1). Fishers targeted a variety of species and the composition of observed sets targeting specific species varied regionally (Table 1). Across all regions mako sharks were most often captured in observed sets targeting swordfish or mixed species (Table 1). That nearly all mako sharks captured in the NED were captured on sets targeting swordfish corresponds to the nearly exclusive targeting of swordfish in observed sets in this region. The SAB and MAB were the only regions with observed sets that specifically targeted sharks, accounting for 7.6% and 13.2% of observed mako shark captures in each region, respectively (Table 1).

Mako sharks were captured on sets with estimated hook depths ranging from 2 m – 110 m, and median set depths increased regionally in a north-south direction, from 14.6 m in the NED to 65.8 m in the GOM (Fig 2). Mako sharks were captured in SST that ranged from 6.0 ℃ – 33.7 ℃, and temperatures were on average warmer in the southern regions (GOM, FEC, and SAB, combined median = 25.5 ℃) compared to the northern regions (MAB, NEC, and NED, combined median temperature = 20.8 ℃; Fig 2). Leader lengths ranged from 0.5 m – 30.8 m, and sharks in the GOM were captured on sets using leaders that were consistently longer (median = 18.3 m) than other regions (combined median = 5.5 m; Fig 2). Mainline length ranged from 3.9 km – 90.7 km, and mainline lengths were on average shorter in the MAB than other regions (Fig 2). Average soak time was relatively consistent between regions at 8.5 h but ranged from 0.4 h – 38.3 h (Fig 2). Captured mako sharks ranged in size (FL) from 60 cm – 480 cm, with an average of 152 cm. Although there was considerable overlap, there was a tendency for captured sharks to be on average larger in the southern regions (GOM, FEC, and SAB, combined median = 180 cm FL) relative to those captured in the MAB, NEC, and NED regions (combined median = 143 cm FL) (Fig 2).

When considering all mako sharks with recorded fate at haul back in the six major regions (n = 7720), AVS was significantly greater in the three northern regions (MAB, NEC, and NED) than the southern GOM, FEC, and SAB regions (Table 2). Model estimated AVS ranged from a high of 0.77 in the NED to ~0.65 in the SAB and GOM (Table 2). Sex was recorded for 6026 mako sharks (F = 2591, M = 3435), and there was no significant difference in AVS between sexes (β = −0.09, 95% CI: −0.21–0.02, Table 3). After removing incomplete records, we retained 4156 mako shark observations for analysis of the effects of fishing and environmental covariates on AVS. Three candidate models were competitive (ΔAIC ≤ 2) and cumulatively accounted for 76% of the combined model weight (Table 4). All variables in the top-supported model were informative (85% CI did not cross 0), and included FL, mainline length, SST, and soak time (Fig 3). The second most supported model excluded FL, and the third model added an uninformative interaction between SST and soak time (Table 4). Consequently, we used the top model for inference. Estimated AUC of the top-model based on LOOCV was 0.58.

Soak time had the greatest effect on AVS (Fig 3), with AVS predicted to decrease by ~15% if soak time increases from 8 h to 12 h when all other variables are held constant (Fig 4). There was a negative effect of SST on AVS (Fig 3), with

**Table 1. Regional differences in target species.**

| Region | Target | Hook depth (m) | % Total sets | % Mako |
|---|---|---|---|---|
| GOM | BET | 36.6 (NA) | 0.02 | 0.00 |
| | DOL | 9.7 (4–11) | 0.24 | 0.10 |
| | MIX | 65 (33–110) | 50.61 | 43.62 |
| | SWO | 47.6 (27–104) | 13.41 | 35.29 |
| | TUN | 76.1 (38–119) | 11.08 | 6.07 |
| | YFT | 74.3 (35–128) | 24.64 | 14.92 |
| FEC | BET | 36.6 (NA) | 0.11 | 0.00 |
| | DOL | 15.9 (7–20) | 0.17 | 0.00 |
| | MIX | 43.4 (26–75) | 19.61 | 28.64 |
| | SWO | 44.4 (24–82) | 77.87 | 70.35 |
| | TUN | 37 (22–49) | 2.19 | 1.01 |
| | YFT | 65.8 (NA) | 0.06 | 0.00 |
| SAB | DOL | 13.1 (4–48) | 7.67 | 7.67 |
| | MIX | 37.2 (9–70) | 31.19 | 32.74 |
| | SHK | 21.9 (NA) | 0.15 | 8.70 |
| | SWO | 39.8 (20-77) | 58.4 | 50.64 |
| | TUN | 33.1 (11-46) | 2.49 | 0.26 |
| | YFT | 38.4 (NA) | 0.10 | 0.00 |
| MAB | BET | 36 (16–40) | 1.77 | 0.43 |
| | DOL | 5.6 (4–7) | 0.92 | 0.12 |
| | MIX | 30 (7–219) | 48.69 | 47.24 |
| | SHK | 8 (2–20) | 1.95 | 13.25 |
| | SWO | 34.4 (16–68) | 12.79 | 28.41 |
| | TUN | 26.9 (4–73) | 29.52 | 9.08 |
| | YFT | 27.5 (11–46) | 4.36 | 1.47 |
| NEC | BET | 26.7 (16–40) | 1.58 | 0.50 |
| | MIX | 26.1 (13–48) | 56.08 | 62.90 |
| | SWO | 27.8 (13–60) | 22.06 | 28.57 |
| | TUN | 22.9 (15–40) | 19.29 | 7.69 |
| | YFT | 40.1 (22–42) | 0.99 | 0.07 |
| NED | MIX | 40.2 (NA) | 0.17 | 0.07 |
| | SWO | 15.1 (9–31) | 99.49 | 99.87 |
| | TUN | 13.7 (13–15) | 0.34 | 0.07 |

Regional differences in target species, mean and range of estimated maximum hook depth, and percent of total shortfin mako shark (*Isurus oxyrinchus*) captures by target species in each region recorded by fisheries observers stationed on U.S. pelagic longline vessels in the Gulf of Mexico (GOM), Florida East Coast (FEC), South Atlantic Bight (SAB), Mid Atlantic Bight (MAB), Northeast Coastal (NEC), and Northeast District (NED) regions of the Atlantic Ocean, 2000–2020. NA range values indicate < 3 sets observed. Targets are bigeye tuna (BET), *Mahi mahi* (DOL), mixed species (MIX), shark (SHK), swordfish (SWO), tuna (TUN), and yellowfin tuna (YFT).

predicted AVS decreasing ~1.5% for each 1 degree increase in SST when other variables are held constant (Fig 4). There were also negative effects of mainline length and shark size on AVS, although these effects were not as strong (Fig 3), with an average decrease in predicted AVS of ~0.2% for each 1 km increase in mainline length and ~0.4% for each 10 cm increase in FL when all other variables are held constant (Fig 4).

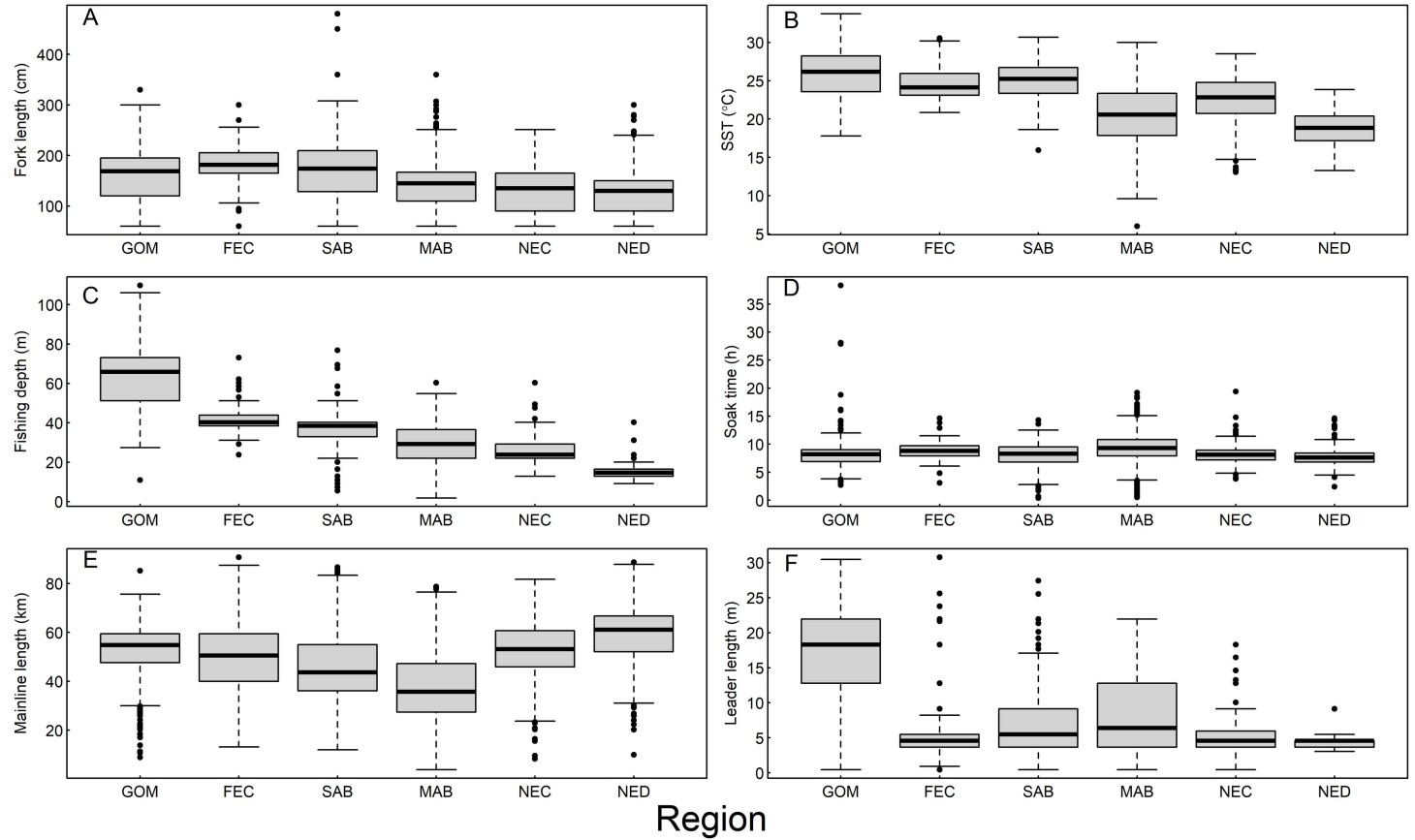

**Fig 2. Regional fishing conditions.** Box and whisker plots illustrating distributions of (A) fork length, and (B) sea surface temperature (SST), (C) estimated maximum hook depth, (D) soak time, (E) mainline length, and (F) leader length of longline sets in which shortfin mako sharks (*Isurus oxyrinchus*) were captured in the Gulf of Mexico (GOM), Florida East Coast (FEC), South Atlantic Bight (SAB), Mid Atlantic Bight (MAB), Northeast Coastal (NEC), and Northeast Distant (NED) regions of the **U.**S. pelagic longline fishery as recorded by fisheries observers, 2000–2020.

**Table 2. Regional at-vessel survival.**

| Region | Alive | Dead | β (95% CI) | AVS (95% CI) |
|---|---|---|---|---|
| Gulf of Mexico | 642 | 330 | – | 0.65 (0.61–0.69) |
| Florida East Coast | 140 | 59 | 0.21 (−0.13–0.55) | 0.69 (0.62–0.76) |
| South Atlantic Bight | 255 | 136 | −0.01 (−0.26–0.25) | 0.64 (0.59–0.70) |
| Mid Atlantic Bight | 2325 | 933 | 0.24 (0.08–0.40) | 0.70 (0.67–0.73) |
| Northeast Coastal | 1036 | 370 | 0.39 (0.20–0.57) | 0.73 (0.70–0.76) |
| Northeast Distant | 1171 | 3332 | 0.62 (0.43–0.81) | 0.77 (0.74–0.80) |

Number of shortfin mako sharks (*Isurus oxyrinchus*) recorded as alive or dead at haulback by fisheries observers on U.S. Atlantic pelagic longline vessels in six monitored fishing regions, 2000–2020, regression coefficients (*β*) and 95% confidence intervals (CI) of the categorical effect of fishing area on at-vessel survival (AVS) probability (reference category = Gulf of Mexico) based on a mixed effects logistic regression model, and model estimated AVS.

## Post-release survival

Thirty-one mako sharks were tagged with PAT devices between April 2022 – May 2024, of which 27 tags deployed on sharks ranging from 91–270 cm FL transmitted data (Table 5). All sharks were tagged off PLL vessels operating in the

**Table 3. Sex-based at-vessel survival.**

| Sex | Alive | Dead | AVS | 95% CI |
|---|---|---|---|---|
| Female | 1773 | 818 | 0.62 | 0.53–0.69 |
| Male | 2280 | 1155 | 0.59 | 0.51–0.67 |

Number of female and male shortfin mako sharks (*Isurus oxyrinchus*) recorded as alive or dead at haul back by fisheries observers on U.S. Atlantic pelagic longline vessels, 2000–2020, and estimated at-vessel survival (AVS) and 95% confidence intervals (CI) based on a mixed effect logistic regression of the effect of sex on AVS.

**Table 4. At-vessel survival model selection results.**

| Model | K | ΔAIC | $w_i$ |
|---|---|---|---|
| FL+ML+SST+ST | 6 | 0.00 | 0.38 |
| ML+SST+ST | 5 | 1.23 | 0.21 |
| FL+ML+SST+ST+SST*ST | 7 | 1.64 | 0.17 |
| ML+SST+ST+SST*ST | 6 | 2.80 | 0.10 |
| FD+LL+FL+ML+SST+ST | 8 | 3.92 | 0.05 |
| FD+LL+FL+ML+SST+ST+SST*ST | 9 | 5.50 | 0.02 |
| FD+LL+ML+SST+ST+SST*ST | 8 | 6.72 | 0.01 |
| FL+SST+ST | 5 | 6.97 | 0.01 |
| SST+ST | 4 | 8.09 | 0.01 |
| FD+LL+FL+ML+SST+ST+HD*SST+FL*ST+SST*ST | 11 | 8.69 | 0.01 |
| FD+FL+SST+ST | 6 | 8.87 | 0.00 |
| FL+SST+ST+SST*ST | 6 | 8.91 | 0.00 |
| LL+SST+ST | 5 | 9.91 | 0.00 |
| SST+ST+SST*ST | 5 | 10.00 | 0.00 |
| FD+SST+ST | 5 | 10.08 | 0.00 |
| FD+FL+SST+ST+FD*SST | 7 | 10.50 | 0.00 |
| FD+FL+SST+ST+SST*ST | 7 | 10.78 | 0.00 |
| FD+LL+FL+SST+ST | 7 | 10.80 | 0.00 |
| LL+SST+ST+SST*ST | 6 | 11.77 | 0.00 |
| FD+LL+SST+ST | 6 | 11.89 | 0.00 |
| FD+SST+ST+SST*ST | 6 | 11.97 | 0.00 |
| FD+LL+FL+SST+ST+SST*ST | 8 | 12.69 | 0.00 |
| FD+FL+SST+ST+FD*SST+FL*ST+SST*ST | 9 | 14.32 | 0.00 |
| FD+FL+FD*FL | 5 | 14.53 | 0.00 |
| SST | 3 | 22.83 | 0.00 |
| FD+FL+SST | 5 | 23.68 | 0.00 |
| FD+SST | 4 | 24.67 | 0.00 |
| FD+SST+HD*SST | 5 | 25.44 | 0.00 |
| ST | 3 | 27.83 | 0.00 |
| FD+FL | 4 | 30.02 | 0.00 |
| FL | 3 | 30.59 | 0.00 |
| ML | 3 | 32.71 | 0.00 |
| FD | 3 | 32.92 | 0.00 |
| Null | 2 | 35.31 | 0.00 |
| LL | 3 | 35.41 | 0.00 |

Model selection results including the number of parameters (K), difference in Akaike information criteria between each model and the top supported model (ΔAIC) and Akaike weights (wi) for 35 mixed effects logistic candidate models of shortfin mako shark (*Isurus oxyrinchus*) at-vessel survival in the U.S. Atlantic pelagic longline fishery based on Pelagic Observer Program data, 2000–2020. Model covariates are fork length (FL), mainline length (ML), sea surface temperature (SST), soak time (ST), maximum estimated fishing depth (FD), and leader length (LL).

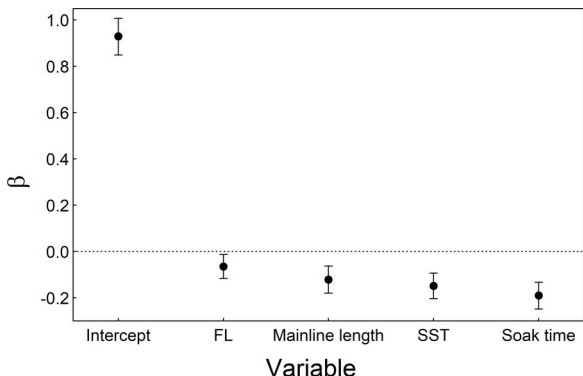

**Fig 3. At-vessel survival coefficients.** Regression coefficients (β) and 85% confidence intervals of the estimated fixed effects of fork length (FL, cm), mainline length (km), sea surface temperature (SST, ℃), soak time (h) and the intercept from the top supported mixed effects logistic regression model of shortfin mako shark (*Isurus oxyrinchus*) at-vessel survival in the U.S. pelagic longline fishery based on pelagic observer program observations, 2000–2020. All variables were scaled and centered.

MAB (Fig 1) between North Carolina and New York, USA. Of sharks that provided data, 19 survived the full 28-day deployment period, 5 experienced pre-mature tag detachment (tracking duration range: 3–23 days), and we detected 3 mortalities (Table 5). Mortalities occurred at 10-, 14-, and 23-days post-deployment (Table 5). The mortality event that occurred at 14-days was attributed to tag ingestion by an endothermic predator. Starting on day 14 the diving pattern changed, and although the tag regularly recorded depths > 400 m, the temperature remained within a narrow range of 22.5 °C – 26.9 °C until the tag began transmitting on day 21 (Fig 5). This temperature range is consistent with ingestion by a lamnid shark, such as a white shark (*Carcharodon carcharias*) or larger mako shark [51,52]. The other two mortalities were identified by the tag rapidly descending to 1800 m and triggering the emergency depth release (Fig 5).

There were no obvious relationships between capture conditions and mortality events. All tagged sharks were hooked in the mouth with circle hooks, no serious injuries were noted, all leaders were cut with ≤ 1 m remaining (Table 5), and the hook was removed from only one shark, which survived. The three mako sharks that died all swam away strongly upon release, were captured in a range of SST's (15 °C – 26.6 °C), in relatively shallow sets (33 m – 57 m), and with soak times ranging from 2–9.5 hours (Table 5). Two of 3 sharks that died were classified as lethargic at haulback, but we found no significant relationship between haulback condition and mortality ($p = 0.18$). The three MCMC chains of the known-fate model mixed well (R-hat = 0.99). Daily survival probability was estimated at 0.995 (95% credible interval: 0.989–0.998), resulting in a PRS estimate to 28-days of 0.87 (95% credible interval: 0.74–0.93).

### Bycatch survival

Monte Carlo simulations combining our estimates of AVS and PRS suggested regional variability in bycatch survival (Fig 6). Estimated bycatch survival distributions formed two noticeable clusters (Fig 6), with survival generally being greater in the northern regions of the NED (BS = 0.64, 95% CI: 0.53–0.70), NEC (BS = 0.61, 95% CI: 0.52–0.68), and MAB (BS = 0.63, 95% CI: 0.52–0.69) than the southern regions of the SAB (BS = 0.59, 95% CI: 0.50–0.65), FEC (BS = 0.57, 95% CI: 0.49–0.64) and GOM (BS = 0.58, 95% CI: 0.49–0.64).

### Discussion

Quantifying the survival of sharks captured as bycatch in commercial fisheries is vital for effective stock assessment and for informing conservation measures such as retention bans. By integrating AVS based on fisheries observer data with PRS based on satellite telemetry we estimated that regionally mean mako shark bycatch survival in the U.S. Atlantic

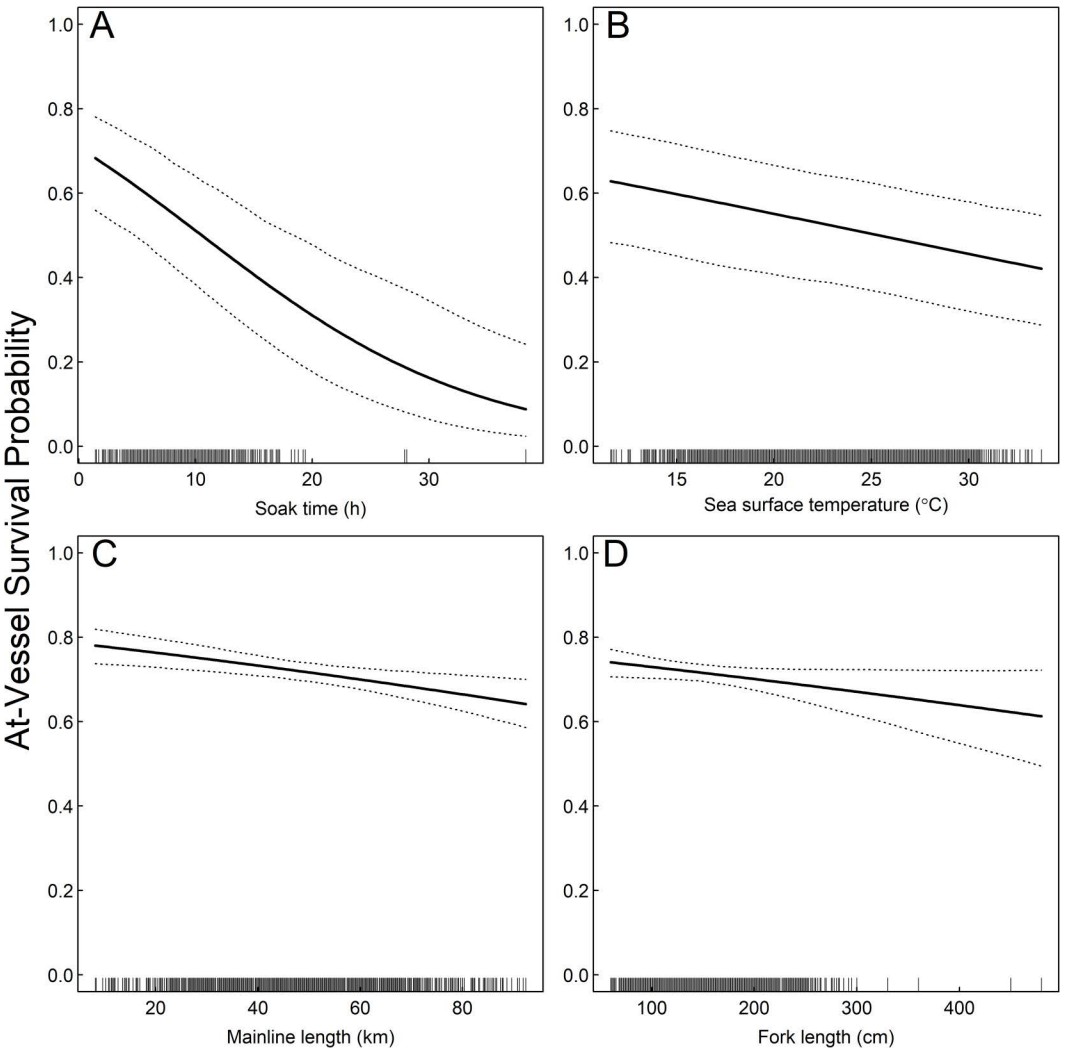

**Fig 4. Covariate effects on AVS: Predicted At-vessel survival probability of shortfin mako sharks (*Isurus oxyrinchus*) in the U. S. Atlantic pelagic longline fishery as a function of (A) soak time (h), (B) sea surface temperature (°C), (C) mainline length (km), and (D) fork length (cm).** Predictions made from top performing mixed effects logistic regression model and hold all other variables constant at their mean values. Dashed lines represent 95% confidence intervals. Rug plots illustrate distribution of data.

PLL fishery ranged from 0.57 to 0.64 (Fig 6). Our estimates are greater than those reported in similar studies in the South Pacific [0.49,[26]] and Canadian Atlantic fishery [0.51, [15]]. A slow growth rate and late age of maturity [18 years for females, [50]] combined with low productivity result in naturally slow population growth rates for the species, making mako sharks particularly susceptible to overfishing [4]. Consequently, even relatively low levels of fishing mortality can significantly reduce population growth [5]. Recent projections suggested that even in the absence of any fishing mortality, the North Atlantic mako shark stock is expected to continue to decline until 2035, and suggest that a total allowable catch ≤ 300 t (inclusive of bycatch mortality) is necessary to rebuild the stock to a sustainable level by 2070 with a ≥ 60% probability [53]. However, reported catches in the North Atlantic from 2015 to 2017 averaged approximately 3,100 t annually—more than ten times the recommended limit [53]. Although mako shark bycatch survival in the U.S. PLL fishery is relatively high compared to other PLL fisheries, our results suggest that on average 36% − 43% of hooked mako sharks

**Table 5. Satellite tagged mako shark data.**

| Deployment date | DAL | Fate | Fork length | Landing condition | Release condition | Leader | SST | Soak time | Hook depth |
|---|---|---|---|---|---|---|---|---|---|
| 2022-04-12 | 28 | A | 150 | Lethargic | Weak | 0.61 | 14.8 | 3.5 | 33 |
| 2022-04-13 | 28 | A | 180 | Active | Strong | 0.91 | 14.3 | 2.0 | 33 |
| 2022-04-20 | 14 | D | 150 | Lethargic | Strong | 0.30 | 18.3 | 9.5 | 57 |
| 2022-04-21 | 28 | A | 150 | Lethargic | Strong | 0.30 | 19.1 | 9.0 | 57 |
| 2022-04-22 | 28 | A | 150 | Active | Strong | 0.30 | 14.3 | 2.0 | 38 |
| 2022-04-23 | 23 | D | 210 | Active | Strong | 0.30 | 15.0 | 3.5 | 38 |
| 2022-04-30 | 6 | A | 150 | Lethargic | Strong | 0.61 | 12.9 | 5.0 | 201 |
| 2022-04-30 | 28 | A | 150 | Lethargic | Weak | 0.30 | 13.1 | 6.75 | 201 |
| 2022-07-24 | 3 | A | 120 | Active | Strong | 0.46 | 24.7 | 11.0 | 40 |
| 2022-08-15 | 28 | A | – | – | – |  | – | – | – |
| 2022-09-12 | 28 | A | 240 | Active | Strong | 0.91 | 26.3 | 10.45 | 26 |
| 2022-10-11 | 10 | D | 120 | Lethargic | Strong | 0.30 | 26.6 | 2.0 | 33 |
| 2022-10-11 | 28 | A | 120 | Active | Strong | 0.30 | 26.2 | 2.0 | 33 |
| 2022-10-12 | 28 | A | 120 | Active | Strong | 0.61 | 23.8 | 17.0 | 27 |
| 2022-10-12 | 28 | A | 120 | Active | Strong | 0.30 | 22.3 | 9.3 | 27 |
| 2022-11-05 | 28 | A | 112 | Lethargic | Weak | 0.30 | 19.9 | 12.0 | 150 |
| 2022-11-06 | 28 | A | 150 | Active | Strong | 0.61 | 19.4 | 8.0 | 22 |
| 2022-12-07 | 28 | A | – | Active | Strong | 0 | 27.0 | 4.0 | 998 |
| 2022-12-28 | 15 | A | 122 | Active | Strong | 0 | 22.9 | 8.0 | 1200 |
| 2022-12-29 | 28 | A | 213 | Active | Strong | 0.15 | 23.9 | 4.0 | 998 |
| 2023-01-28 | 23 | A | 91 | Active | Strong | 0.30 | 23.7 | 5.0 | 600 |
| 2023-01-28 | 28 | A | 121 | Active | Strong | 0.30 | 23.5 | 4.0 | 600 |
| 2023-08-13 | 28 | A | 180 | Active | Strong | 0.76 | 24.3 | 7.4 | 27 |
| 2023-09-05 | 28 | A | 180 | Active | Strong | 0.91 | 22.5 | 5.0 | 38 |
| 2023-10-28 | 12 | A | 150 | Active | Strong | 0.02 | 21.2 | 6.1 | 38 |
| 2023-12-04 | 28 | A | 150 | Active | Strong | 0 | 15.4 | 4.3 | 38 |
| 2024-05-05 | 28 | A | 270 | Active | Strong | 0.15 | 23.9 | 7.0 | 37 |

Summary information for shortfin mako sharks (*Isurus oxyrinchus*) tagged with pop-up archival tags on U.S. pelagic longline vessels in the Atlantic Ocean, 2022–2023 that transmitted data. Data include the days at liberty (DAL), fate as of the last day with tracking data (A = alive, D = dead), fork length (cm), landing and release condition, length of leader material remaining (m), sea surface temperature (SST; ℃), soak time (h), and approximate hook depth (m).

may still die depending on location. Traditionally U.S. landings account for a relatively small portion of the total reported landings in the North Atlantic, ~11% between 2010–2016 [54]. Considering the North Atlantic stock as a whole, if bycatch survival is lower in other regions or for other PLL fleets relative to the US, particularly fleets that account for larger proportions of total landings (e.g., Spain -~47%, Portugal – 20%, Morocco – 16%) [54], then given the life history characteristics of the species, stock recovery may be slower than desired in the presence of retention bans if large numbers of mako sharks are still hooked as bycatch.

## At-vessel survival

The contribution of the two primary components of overall bycatch survival, AVS and PRS, may not be equal. We found that mako shark AVS was considerably lower than PRS in the U.S. PLL fishery, suggesting mortality events are more likely to occur before the shark is brought to the boat. Our estimate of mean AVS in the northern MAB (0.70), NEC (0.73) and NED (0.77) regions were similar to other recent studies in the western Atlantic, which ranged from 0.74 in the Canadian

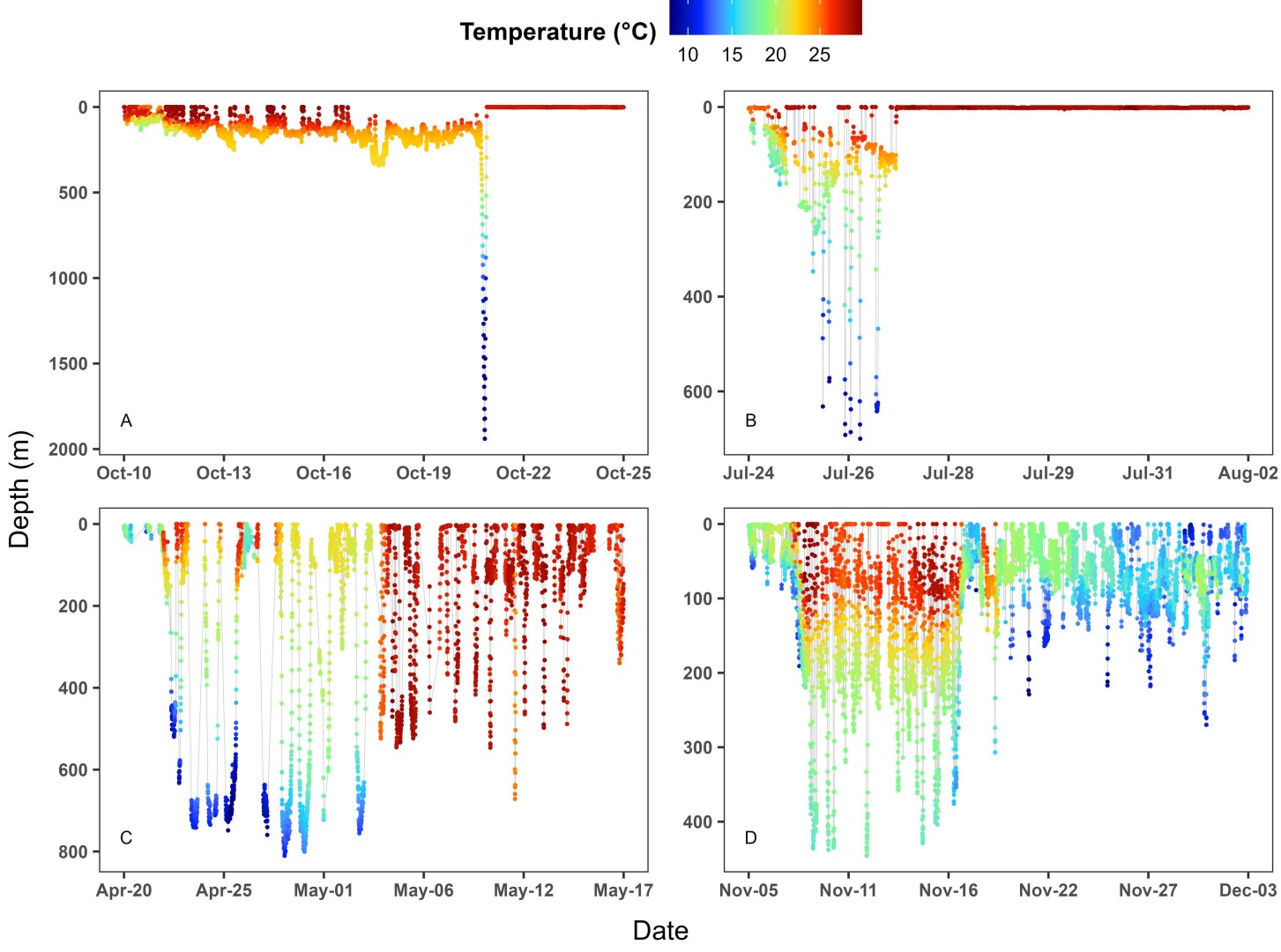

**Fig 5. Depth-temperature plots.** Representative depth-temperature plots of shortfin mako sharks (*Isurus oxyrinchus*) tagged with pop-up satellite tags hooked on U.S. pelagic longline vessels in the western North Atlantic Ocean illustrating a mortality event in which the shark sank below 1800 m and triggered the depth release mechanism **(A)**, a pre-mature tag detachment **(B)**, a tag ingested by an endothermic animal **(C)**, and a shark that survived the full 28-day deployment period **(D)**.

PLL fishery [15] to 0.71 across the entire U.S. Atlantic fleet based on earlier POP observations [17]. Within the U.S. PLL fishery we found mean AVS was lower in the southern SAB (0.64), FEC (0.69), and GOM (0.65) regions, and were similar to that reported for the Portuguese PLL fishery [16]. Gallagher et al. [17] performed a similar study using POP data collected during 1995–2012 and, contrary to our results, found no significant effects of any environmental, biological, or fishing-related variables on mako shark AVS. An explanation for the different results is not readily apparent, although it is plausible that the data filtering used and shorter time period considered resulted in a smaller sample size that might have reduced the power to detect significant relationships.

Our finding of regional differences in AVS within the U.S. PLL fleet illustrates the importance of accounting for spatial variation in survival studies, even within the same national fleet. Environmental differences, specifically water temperature, may at least partially explain why AVS varied between regions. We found a significant effect of SST on

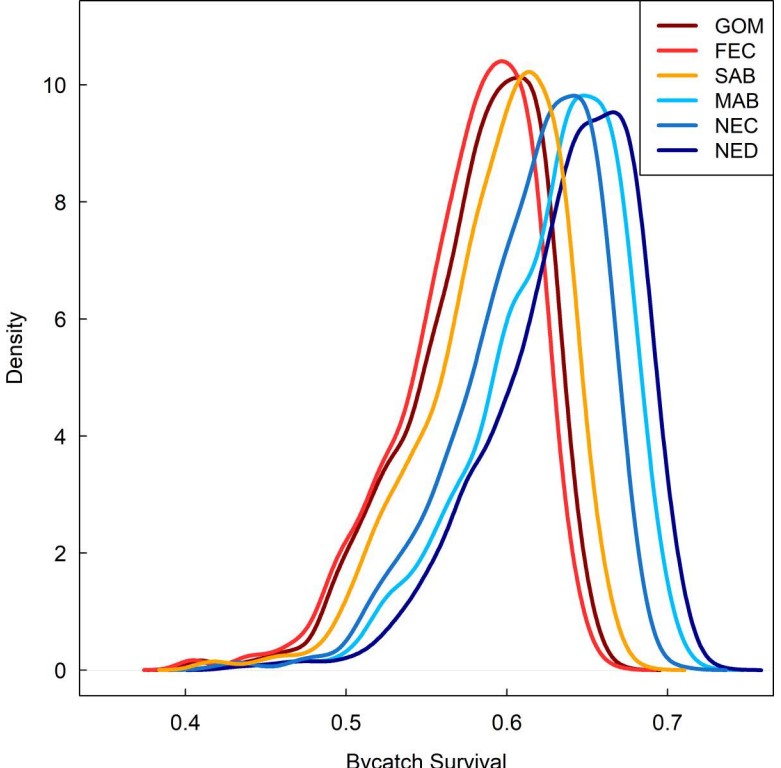

**Fig 6. Bycatch survival estimates.** Distributions of estimated bycatch survival of shortfin mako sharks (Isurus oxyrinchus) hooked on U.S. pelagic longline vessels in the Gulf of Mexico (GOM), Florida East Coast (FEC), South Atlantic Bight (SAB), Mid Atlantic Bight (MAB), Northeast Coastal (NEC), and Northeast Distant (NED) regions of the Atlantic Ocean. Distributions represent proportion of sharks surviving the capture and release process from 1000 Monte Carlo simulations incorporating modeled estimates of at-vessel and post-release survival, and regional-specific fishing conditions.

AVS, with survival decreasing as temperature increased. The relationship between elevated water temperature and stress response has been documented in various species [e.g., [36–38]]. Any deviation from the optimal thermal range for mako sharks could impose additional metabolic stress potentially impacting AVS and as regional endotherms [39] mako sharks may be particularly susceptible to overheating when struggling on a line in warm temperatures for long periods. The latitudinal trend in AVS corresponded with an inverse trend in SST (Fig 2). Mako shark AVS was greater in the northern regions compared to the southern regions where mako sharks were hooked in waters with consistently warmer SST. Thus, higher temperatures in the GOM, FEC, and SAB regions may have conferred a higher baseline mortality than in the cooler regions.

Exhaustion associated with increased time on the line is known to affect the severity of acidosis in sharks, where blood lactate concentration increases as capture duration increases [55–57], which can subsequently cause mortality [25,58]. Massey et al. [59] found a negative correlation between time on the line and AVS of mako and blue sharks (*Prionace glauca*) caught on PLL in the South Pacific Ocean, and similar results have been found for other species [58,60]. While POP observers could not measure time on the line, soak time may serve as a proxy measure as longer soak times increase the time possible for a shark to remain hooked. Of the variables considered, we found soak time had the greatest influence on AVS, with AVS decreasing precipitously as soak time increased such that the probability of survival when soak time is 4 hours is 27% greater than 12 hours, all else being equal (Fig 4). Epperly et al. [61] also found a negative

effect of soak time on mako shark AVS by U.S. PLL fishers on the Grand Banks during 2002–2003. Carruthers et al. [35] found a negative but non-significant effect of soak time on AVS in the Canadian PLL fishery during 2001–2004. The range of soak times in which mako sharks were captured was not reported by Carruthers et al. [35], although soak times appeared to be, on average, longer (~13 hours) than the present study, and it is plausible that the range of times makos were captured was not sufficient for a significant effect to be detected. Both blue and porbeagle shark (*Lamna nasus*) AVS was found to significantly decrease as soak time increased [35].

We found a significant although not particularly strong negative correlation between AVS and mainline length (Fig 4). It is plausible that this relationship may be a consequence of increased retrieval times and distance a hooked shark is dragged through the water as mainline length increases. Other studies have found that large mako sharks have greater AVS than smaller individuals [16,35]. Our model results suggested a negative effect of size on survival in the U.S. PLL fishery, however we note that these estimates may be biased by a few large individuals and given the relatively small effect size, we hesitate to interpret our results as representative of a strong biological effect.

The predictive success of our AVS model at 59% suggests there are likely other factors we did not measure that predict AVS. Direct measure of time on the line rather than soak time may increase model performance, although we note the time on the line cannot be controlled by the fisher and cannot be directly managed, whereas soak time can. We used SST as an environmental measure; however, water temperature at hook depth may be more informative in future analyses as sharks spend most of their time at that temperature while hooked and temperature at depth may not directly correlate with SST. Our measure of hook depth represents a single estimate of maximum hook depth applied across an entire set. Hook depth likely varies throughout a set due to interacting factors such as drift speed, weight used, and distance between floats. Consequently, more fine-scale measures of individual hook depth may be able to detect any potential effects of hooking depth. Sharks tagged for PRS estimation during 2022–2024 were captured within a larger depth range than the maximum estimated fishing depth reported by observers during 2000–2020, with estimated hook depths ranging from 22 – 1200m and 18% of tagged makos captured at hook depths ≥ 600 m (Table 5). This difference likely reflects a recent change in fishing tactics towards deeper sets in the MAB, especially for fishers targeting bigeye tuna (*Thunnus obesus*), and currently, it is unclear how capture on deep sets affect AVS.

## Post-release survival

We found mako shark PRS (0.87) was relatively high, suggesting mako sharks released alive have a high probability of surviving. Our estimates are similar to recent findings in Pacific PLL fisheries, including the tuna PLL fishery in the southwest Pacific Ocean [30-day PRS = 0.90; [26]] and the Hawaiian deep-set fishery [94%; [31]], but greater than estimates reported from studies in the Atlantic Ocean. Estimates in the Canadian PLL fleet have consistently averaged ~0.70 [15,19], whereas Miller et al. [18] reported 30-day PRS as 0.77 based on PATs deployed throughout the Atlantic in ongoing work conducted by ICCAT. Bowlby et al. [20] combined data from several datasets of PATs deployed on PLL vessels between 2011–2019 in the Atlantic Ocean and estimated PRS ~ 0.66. Data from mako sharks captured on U.S. PLL vessels is absent or poorly represented in these previous studies in the Atlantic, suggesting potential regional and fleet-specific variation in PRS.

There are several plausible reasons for the greater PRS we observed relative to previous studies in the Atlantic Ocean. Francis et al. [26] found that PRS decreased as the length of the leader remaining relative to shark size increased, and the length of the trailing leader for all sharks in the present study was ≤ 1 m. All mako sharks tagged in the present study were hooked in the mouth, likely because of the use of circle hooks, which significantly reduce the probability of gut or foul hooking [61,62]. Gut-hooked mako sharks have lower PRS than mouth-hooked individuals [20], likely because of internal injuries associated with gut-hooking [63], which appears to be mitigated by the adoption of circle hooks. Injured mako sharks have lower survival than uninjured individuals [15,26], and the fact that observers did not record any noticeable external injuries to tagged mako sharks may also contribute to the high PRS we observed. Potential for injury can

be mitigated by using circle hooks, as well as leaving sharks in the water, which is standard practice given the requirements to release all live mako sharks in a manner that causes the least harm, and the logistical difficulties associated with handling large individuals. Thus, the use of circle hooks and practices of cutting the leader while the shark remains in the water so that < 1 m of trailing leader remains, which are predicted to maximize pelagic shark PRS [31,64], likely synergize resulting in relatively high PRS in the U.S. PLL fishery. However, despite best practices, some portions of hooked mako sharks must sustain injuries and their survival is likely reduced. Additionally, it is plausible that some fishers may handle sharks differently when POP observers are not present. As such, we suggest that our estimates be interpreted cautiously until larger sample sizes are available, or they are replicated in future fishery studies.

Most pelagic shark post-release mortalities occur within five days of release and often within 48 hours [64], and this pattern is consistent in mako sharks [15,18,20,26]. Our study is unique in that all three mortalities occurred ≥ 10 days post release suggesting all tagged sharks could recover from acute capture stress, including those classified as lethargic on haulback and weak on release. Potential reasons for mortality events beyond the initial release period include complications from injuries, embedded hooks, or trailing gear that result in physiological stress, prevent or impede foraging, or increase susceptibility to predation [31,63–65]. Because most fishing-related mortalities occur quickly after release, a 30-day tracking period is often considered appropriate to quantify PRS while minimizing costs and the probability of PAT tag malfunction and maximizing the likelihood of successful data transmission [64]. This was our rationale for using tags with a maximum 28-day deployment period. Other studies have observed mortalities, which authors attributed as fishing-related up to 50 days post-release [20,26], and it is plausible that additional mortalities of makos tagged in this study may have occurred beyond the 28-day deployment period. However, as time since release increases, the ability to differentiate between fishing-related and natural mortalities becomes more difficult.

With only three observed mortalities we were not able to directly model the effects of covariates on PRS. Although two sharks that died were classified as lethargic when landed, we found no statistical relationship between condition at haulback and survival, and four sharks classified as lethargic survived to 28 days. Additionally, all sharks that died were classified as strong on release, and all those classified as weak survived (Table 5). These results are similar to those reported for mako sharks captured on sportfishing gear, in which two of three sharks that died within 30 days of release swam away well after release, and all sharks classified as moribund on release except for one that exhibited severe bleeding survived [56]. Unless severely injured, predicting mako shark PRS based on perceived visual condition during capture and release seems unreliable [18,56].

As all tagging occurred in the MAB, we necessarily assumed consistent fleet wide PRS to estimate bycatch survival in all regions. However, given that AVS differs regionally, PRS may as well. It has been speculated that shark condition at haul back resulting from conditions experienced while on the hook is likely to have a greater influence on shark PRS than handling practices during release [24]. If so, it could be expected that environmental conditions and fishing practices that influence AVS similarly influence PRS, and consequently, regional PRS may mirror AVS trends. If true, then our bycatch survival estimates of mako sharks may be overestimated in regions like the GOM and underestimated in the NED. Future efforts focused on quantifying PRS across a wider spatial sale would provide clarity in this regard.

## Management implications

Our research shows that although mako sharks are expected to have high survival rates when released alive in the U.S. Atlantic PLL fishery, most mortalities are likely to occur before the shark is brought to the vessel. Bycatch survival estimated from combined AVS and PRS rates varied regionally in the U.S. PLL fishery and may be low enough to result in sufficient fishing mortality, even in the presence of a retention ban, to slow Atlantic mako shark stock recovery below desired levels if large numbers of mako sharks are encountered as bycatch. The high PRS we observed suggests that practices including not removing sharks from the water, using circle hooks, and cutting leaders as close to the hook as possible are associated with very high survival probabilities. Given that at-vessel mortality is the larger mortality component, actions to increase AVS, such as reducing soak times, could increase bycatch survival. However, following the

principles of the sequential mitigation hierarchy [66] the most effective strategy is likely to pair retention bans with management actions that reduce the overall incidence of mako shark bycatch in the first place. O'Farrell and Babcock [6] illustrate how this could be accomplished by a combination of dynamic time-area closures and limitations on the use of light sticks in the northern regions of the U.S. PLL fishery, which is where the greatest amount of mako shark bycatch occurs (Fig 1). Given that U.S. mako shark catches represents a relatively small proportion of the total North Atlantic catch [54], any action taken by the U.S. fleet alone—without broader international adoption—is unlikely to provide meaningful benefits to the stock. Continued research to quantify bycatch survival in different regions and for different PLL fleets in the Atlantic Ocean, as in the present study and previously estimated for the Canadian fleet [15], will aid future stock assessments by providing tools to estimate total fishing mortality more accurately.

## Supporting information

**S1 File. Shortfin mako shark tagging data.** Deployment dates, fates, and capture data recorded for all shortfin mako sharks tagged with pop-up archival tags on U.S. pelagic longline vessels in the Atlantic Ocean, 2022–2024. (CSV)

**S2 File. PSAT data.** Complete depth and temperature data from all pop-up archival tags deployed on shortfin mako sharks on U.S. pelagic longline vessels in the Atlantic Ocean during 2022–2024 that transmitted data. (CSV)

## Acknowledgments

We thank W.B. Driggers III for logistical support and valuable input during project initiation. We thank the fisheries observers and cooperating captains who helped in tagging and the staff of the Southeast Pelagic Observer Program at the National Marine Fisheries Service, NOAA, including C. Rewis and S. Cushner, who helped coordinate tagging and provided observer data. The scientific results and conclusions are those of the authors and do not reflect those of any government agency.

## Author contributions

**Conceptualization:** Eric R. Hoffmayer, James A. Sulikowski, Michael E. Byrne.

**Data curation:** Mischa Schultz.

**Formal analysis:** Mischa Schultz, Michael E. Byrne.

**Funding acquisition:** Michael E. Byrne.

**Investigation:** Mischa Schultz.

**Methodology:** Mischa Schultz, Eric R. Hoffmayer, James A. Sulikowski, Michael E. Byrne.

**Project administration:** Michael E. Byrne.

**Resources:** James A. Sulikowski.

**Visualization:** Michael E. Byrne.

**Writing – original draft:** Mischa Schultz, Michael E. Byrne.

**Writing – review & editing:** Mischa Schultz, Eric R. Hoffmayer, James A. Sulikowski, Michael E. Byrne.

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
