## [Decision Letter · Decision Letter 0]

18 Feb 2025

*Isurus oxyrinchus*

Dear Dr. Byrne,

Thank you for submitting your manuscript to PLOS ONE. After careful consideration, we feel that it has merit but does not fully meet PLOS ONE’s publication criteria as it currently stands. Therefore, we invite you to submit a revised version of the manuscript that addresses the points raised during the review process.

The paper has now been reviewed by 3 reviewers. All were supportive of publication, and have provided several corrections and comments that require correction. As such, I am recommending major revisions. I invite you to address the points raised by the reviewers, and to submit a revised version of the manuscript, so it can be further considered for publication.

We look forward to receiving your revised manuscript.

Kind regards,

Rui Coelho, PhD

Academic Editor

PLOS ONE

“We thank W.B. Driggers III for logistical support and valuable input during project initiation. We thank the fisheries observers and cooperating captains who helped in tagging and the staff of the Southeast Pelagic Observer Program at the National Marine Fisheries Service, NOAA, including C. Rewis and S. Cushner, who helped coordinate tagging and provided observer data. NOAA funded this research through the Bycatch Reduction Engineering Program (grant # NA19NMF4720227).  M.E. Byrne was supported by the National Institute of Food and Agriculture, U.S. Department of Agriculture, McIntire-Stennis project number 7006325. This research was not endorsed by the National Marine Fisheries Service. The scientific results and conclusions are those of the authors and do not reflect those of any government agency.”

“MEB - Quantifying and reducing post-release mortality of shortfin mako sharks (Isurus oxyrinchus) captured as bycatch in pelagic long-line fisheries.

Grant number: NA19NMF4720227

Funding agency: National Oceanographic and Atmospheric Administration (NOAA).

Funding agency website: https://www.noaa.gov/.

The funder for this study had no role in study design, data collection and analysis, decision to publish, or preparation of the manuscript.”

Additional Editor Comments:

Dear Authors,

Thank you for submitting the manuscript “Bycatch survival of shortfin mako sharks Isurus oxyrinchus in the U.S. Atlantic pelagic longline fishery” for consideration to publication in PlosOne.

This is an interesting paper that presents important information and results on the discard survival of shortfin mako shark, including at vessel mortality recorded by fishery observers and port-release survival recorded by satellite tagging. Those data and results are usually poorly known on shark species and are important when conducting stock assessments. So overall the paper presents important and useful results that are important to be published in the scientific literature.

The paper has now been reviewed by 3 reviewers. All were supportive of publication, and have provided several corrections and comments that require correction. As such, I am recommending major revisions. I invite you to address the points raised by the reviewers, and to submit a revised version of the manuscript, so it can be further considered for publication.

Best regards,

Rui Coelho

Reviewers' comments:

Reviewer's Responses to Questions

**Comments to the Author**

1. Is the manuscript technically sound, and do the data support the conclusions?

Reviewer #1: Partly

Reviewer #2: Partly

Reviewer #3: Yes

2. Has the statistical analysis been performed appropriately and rigorously?

Reviewer #1: Yes

Reviewer #2: No

Reviewer #3: Yes

3. Have the authors made all data underlying the findings in their manuscript fully available?

Reviewer #1: Yes

Reviewer #2: No

Reviewer #3: Yes

4. Is the manuscript presented in an intelligible fashion and written in standard English?

Reviewer #1: Yes

Reviewer #2: No

Reviewer #3: Yes

Reviewer #1: Review: Bycatch survival of shortfin mako sharks (Isurus oxyrinchus) in the U.S. Atlantic pelagic longline fishery

This paper presents a comprehensive assessment of AVS and PRS for shortfin mako shark from US fleets in the North Atlantic. It is well-written and comprehensive, and I particularly liked the approach of using MC simulation to produce overall bycatch mortality rates. I have no concerns about research ethics. There are two changes that I think are required before the paper is suitable for publication: (1) further detail and potential changes to the analyses in relation to hook depth information, and (2) consideration of bycatch rates predicted from only the significant predictors in the GAMM. My detailed comments are provided in the attached file.

Reviewer #2: This study presents the results of an important data collection on the survival of mako sharks caught by commercial pelagic longline fisheries (US) in the Northwest Atlantic. In this study, the authors assess the survival (i.e. at-vessel survival AVS and post-release survival PRS) of the mako shark as a bycatch specie in the LL fisheries and identify the effects of environmental, biological and fishing practices variables on at-vessel survival. Given their results, the authors discuss the retention ban as a bycatch mitigation measure to the restoration of the abundance level of overfished population.

The manuscript is interesting because it provides important and much-needed data on total mortality (at-vessel and post-release survival) in a species that is highly vulnerable to pelagic longline fisheries, providing a more complete analysis of the impact of this fishery on mako sharks. It also tackles the issue of pelagic sharks’ fishing mortality in a new way through a study focusing on a comparison between two adjacent fishing regions of the north-western Atlantic.

However, the manuscript lacks scientific rigor at several levels:

The manuscript requires improvement in its use of standard English and scientific writing. Some sentences are not complete. (See my detailed comments below for specific examples).

General connections and shortcuts are made between different notions of fisheries, without the underlying logical links being clearly explained (e.g. stock assessment, risk assessment, overall mortality, retention ban...).

The manuscript lacks consistency in the use of the terms “mortality” and “survival”. I think the authors should choose one or the other, but not both as it becomes confusing to follow the logical reasoning.

There is a lack of information on the characteristics of the fishing set observed, in particular their total number, and on gear configuration: the types of hooks used (and in what proportions), the type of bait used, the length of branchlines...

Also, the target species drives fishing practices (day or night) and gear configuration (especially hook depth and bait). This is a very important aspect to consider in this type of study. However, this information is not given (especially the proportions) in the data collected, and above all is not addressed in the discussion.

There is a lack of information on data processing, especially concerning hook depths. What do they correspond to and how were they calculated? The same goes for soak time: is it the soak time of the longline or the soak time of the mainline section between wo floats on which the shark was caught ?

The hypothesis behind the modeling of the effect of each variable on mortality must be indicated and referenced.

Although this is an original and interesting approach carried out with an impressive dataset, it is not clearly explained why the authors compare these two different regions in terms of mako sharks AVS. Also, it is unclear why the authors chose the longitude as the sole criterion to discriminate the two regions.

I suggest the authors include the variable “sex of the animal” in the analysis. I'd advise performing an analysis on the reduced data matrix to assess whether “Sex” could have an effect on AVS - which could have management consequences, especially for populations where individuals of different sex may display different spatial distributions.

As GAMM results show linear relationships in almost all covariates, I would suggest using GLMMs, which would also allow them to know the weight of each covariate on the variability of the dataset. Rerunning the models by selecting only those covariates with a significant effect will increase the level of predictability of the model, which as presented in the study, is rather poor. A careful interpretation of AUC results is needed to present the significance of the selected model.

Therefore, I recommend major revisions with peer-review after corrections.

Please find below my comments on the specific content of the manuscript.

l.59: The authors should define fishing mortality (at-vessel mortality + catch dying on board ? + post-release mortality ?)

l.66-68: “Additionally, the International Union for the Conservation of Nature (IUCN) assessment estimated a ~60% decline in biomass over 75 years [9].” � the authors sould repeat which biomass they are talking about.

l.72: concerning retention ban efficiency, authors should cite Tolotti et al., 2015: “Banning is not enough: The complexities of oceanic shark management by tuna regional fisheries management organizations”

l.75-77 : the authors state something in the introduction that they are trying to show through their study

l.94-96 : missing reference

l.98-103: As I said in my general comments, actually it mainly depends on the target species, for which the depth of capture is adapted. I think it's worth giving other examples of differences in practices that can influence mortality than just depth, such as the type of hook, the type of line, the bait.... There are many references on this subject.

l.105-112: Fig.2: it would be preferable to indicate the CPUE of mako and/or the proportion of mako observed in the total catch rather than just the number of catches, which may conceal very different fishing and/or observation efforts.

l.134-137: why are there so many differences between regions? is it possible to have the respective fishing efforts in each region?

l.137-139: the authors need to justify better than “for analysis purposes”, because these two regions don't have the same observation efforts, the same surface areas covered, and so the environmental variables are surely more variable for WNA than GOM.

It would be interesting to test the distribution of the covariates within the WNA zone to find out how homogeneous it is before grouping these regions together and assuming that the effects aez constant within the region.

l.147: The choice of considering "year" as a random effect is unclear to me in this context. If information on vessel ID is available, it would be more appropriate to retain this covariate, as studies have suggested that variability in survival occurs at the fishing unit level (which summarizes a set of drivers specific to a captain’s fishing practices)

l.202 : missing reference

l.226-228: why these limits? references? or is it the distribution tail of their data? what thresholds have been chosen?

l.246-255 : switch between at-vessel mortality et survival

l.263-264 : “although the effect of neither variable was very strong ». What is a strong effect ?

table 1: I suggest that the authors indicate the coefficients, at least to know whether the effect is positive or negative.

l.317 & 319: pay attention to meaning of the sentence, sharks don’t transmit data

l.342-343: The authors should indicate the type of hook that have been used in the methods part, and in what proportions. The hooking location is essential for survival. It would be nice to show this data for all sharks of the study, not only those that have been tagged, if the data is available of course. Similarly, whether or not sharks leave with the hook is vital information, which, if collected, should be intergrated as a potential mortality factor.

table2: “approximate hook depth”: as indicated above: what does this hook depth correspond to? How was it calculated and at what point in the capture does it correspond?

l.368 : missing reference

l.375 : replace “anonymous” with “ICCAT”

l.376: “the north Atlantic stock is expected to continue to decline ... ” once again, pay attention to the sentence structure: you need to repeat which species you're talking about.

l.377: I suggest that the authors indicate the quantities of mako fished currently and/or before the retention ban, in relation to the 300T TAC. Otherwise it is difficult to appreciate the seriousness of the current situation.

l.381-385: if the authors address the northern Atlantic mako stock, then they have to refer to the landing figures of the other fleets that fish this stock. Also, the last sentence suggests that the retention ban could be a problem, but I'm not sure that's what the authors meant.

l.394: how do the authors interpret these similarities with the results of the portuguese fleet?

l.396-398: if this is the authors’ hypothesis, they can verify it with the data by running the models on the whole dataset, without discriminating the two regions.

l.402: I suggest that the authors link the effect of temperature to that of hook depth. The subject is touched in lines 412 to 414, but not sufficiently developed. The effect of SST on the survival of animals caught at different depths and temperatures is therefore difficult to interpret, unless all individuals were caught above the thermocline, but this would have to be verified with data on hooking depths and thermocline to know.

l.403-404: Given the method used, I don't find it justified to compare the results of driver effects between regions.. The authors looked at the effect of several variables for each region, not the effect of each variable between the two regions.

l.408-409: The effect of SST on survival concerns sharks caught in the WNA, so this result should not be compared with those caught in the GOM. This interpretation would have been correct if the effect had been found for the whole sample.

l.424-427: I think that there are many explanations behind the fishing practices related to target species. This was not mentioned in the introduction or the methods, but it is important to indicate the proportions of fishing sets that targeted this or that species, and perhaps further analysis could integrate the target species as a driver of AVS.

l.463-464: just because the PRS is higher than the AVS does not mean that mako have a better chance of survival after release. I don't find it relevant to compare the two indicators. Otherwise in this case, the authors could compare with other species for which there would not be the same pattern.

l.529: It would be relevant to place bycatch reduction in the context of a sequential mitigation hierarchy of interactions between fishing gear and the non-target resource (Gilman et al., 2023 - Bycatch-neutral fisheries through a sequential mitigation hierarchy - Marine Policy).

Reviewer #3: This is an interesting manuscript that presents an analysis of the survival of mako sharks (Isurus oxyrinchus), based on long-term observations from the Pelagic Observation Programme and telemetry data. The survival of mako sharks caught on longlines remains poorly understood. These sharks often experience high levels of stress and physical injury during capture, which can result in significant mortality. Their high activity levels and aggressive behavior may further increase the risk of injury. Using long-term data and advanced biotelemetry techniques, the authors analyze the survival of this species in two regions. They also propose conditions that could be maintained to improve the survival of mako sharks in longline fisheries.

The authors used data from 6681 observations made in the Western North Atlantic (WNA) and the Gulf of Mexico (GOM). The spatial distribution of the data was not uniform, so the authors correctly divided the material into GOM and WNA. The second source of data was the results of telemetry surveys using pop-up archival tags. This type of tag transmits via satellite the parameters recorded over a pre-determined period of time. In this case, the material was only from the years 2022-2024. The data to determine post-release survival were collected over a limited area. However, the authors selected the area with the highest number of mako shark sightings for this element of the study. Based on the depth and temperature recorded, the authors estimated the medium-term mortality of this species. In conclusion, I believe that the authors have prepared an interesting manuscript, used appropriate data analysis methods and presented the results clearly. In the discussion chapter, I like the caution in making explicit assessments. The authors are aware of the weaknesses of their study. They rightly consider that the soaking time was not the best indicator of the time spent on the line. They also address the smaller sample size in the GOM, stressing that it may not have been sufficient to detect the effect.

Before accepting this manuscript for publication, I suggest that you consider some of the following points.

1. Surprisingly, SST was only a significant factor in increased mako mortality in the colder waters of the WNA. I lack an explanation as to why this factor did not have a significant effect on AVS in the warmer GOM? Or is this an adaptation to local conditions?

2. Why were there regional differences in AVS for mako sharks? What could have been the reasons for the significant differences in the GOM and WNA? The importance of immersion time is particularly surprising. Was this due to differences in depth or some other reason? Does the soak time depend on the depth of the fishery or some other technicality?

3. The case of the mortality of a mako shark on day 14 after release is very interesting. What endothermic predator could have attacked/eaten such a large shark? Could this predator have a temperature of 22.5-26.9 degrees Celsius? What is the accuracy of the temperature readings from the pop-up tags used?

Specific comments:

Lines 131-133: Reference to more recent literature is needed. Publication 29 (Diaz et al. 2009) only gives data up to 2007. Where does it say that after 2007 observer coverage was >8%?

Line 142: What do you mean by hook depth? For those unfamiliar with this fishing technique, an explanation of the term is needed.

Line 187: How was the leader length estimated? Who did it? What was the margin of error? Was this person experienced in this procedure?

Lines 347-350: why did you not describe this analytical procedure in the methods section?

Lines 381-382: Where did you get the data on the US contribution to total landings? References are needed.

Line 478: Information that all PLL fleets used circle hooks should be in the methods section. Information on the type of hook used to catch fish is important when interpreting the data in the context of survival. This is the only place where such information appears, and readers with knowledge of the subject will find it easier to follow the authors' reasoning.

Figures - their quality is poor and should be improved. I think that higher dpi values should be used.

**Do you want your identity to be public for this peer review?** For information about this choice, including consent withdrawal, please see our Privacy Policy

Reviewer #1: **Yes: ** Heather D. Bowlby

Reviewer #2: No

Reviewer #3: **Yes: ** Andrzej Kapusta

---

## [Author Response · Author response to Decision Letter 1]

16 Jun 2025

This is a copy of the "response to reviewers" document

Dr. Coelho,

Thank you for the opportunity to revise our manuscript and apologies for the delayed revision time. We have addressed the comments of the reviewers, and we believe our manuscript is improved as a result. Our responses to specific comments for each reviewer are available below, however we would first like to highlight some of the more general changes. All line numbers reference the line numbers in the document with the track-changes when all mark up in shown in balloons, where revisions are easier to see, unless otherwise noted.

The biggest change is that addressing the concerns of Reviewer 2 led us to completely revise the at-vessel survival analysis. Rather than run 2 separate GAMMs for different regions we used GLMMs and used data from all observer regions, and did not combine 3 regions into a single region (WNA) as we did originally. Our revised approach was as follows:

First, to test for regional differences in AVS we used a GLMM incorporating data from the 6 observer regions with > 100 mako shark observations (NED, NEC, MAB, SAB, FEC, GOM), with region as a categorical fixed effect. This is described in the methods on lines 393-404. We found evidence that AVS was greater in the 3 northern regions (NED, NEC, MAB) compared to the 3 southern regions (SAB, FEC, GOM). Which is similar to our result in the original manuscript, it encompasses more area and allows for more fine scale spatial variability to be observed. The results are presented in Table 2, and in the text on lines 651-654.

Next, to address the possibility of sex-biased survival raised by reviewer 2, before our analysis aimed at determining covariate effects on AVS, we used a GLMM that incorporated all available observations in which the sex of the shark was recorded. We use sex as a categorical fixed effect and found no evidence to suggest a difference in survival between sexes. Given this information, we felt justified in not including sex in our analysis to determine which covariates influenced AVS. This was advantageous because given the large proportion of sharks with no sex data recorded (~23%), including sex in further analyses would have led to a considerable loss of sample size and statistical power. The sex analysis is described in the methods on lines 411-418. The results are presented in Table 3 and on lines 654-656.

Finally, we pooled all observations for the analysis of covariate effects on AVS. We used logistic GLMMs for this analysis instead of a GAMM, as Reviewer 2 suggested. An advantage of this approach relative to our original analyses was that we could construct candidate models made up of different combinations of covariates and rank their support using AIC. The most supported model included only informative (significant) variables, addressing concerns of reviewers 1 and 2 in regards to making predictions from models with uninformative parameters. We found significant negative effects of soak time, SST, mainline length, and shark size on AVS, although the effects of soak time and SST were considerably stronger than mainline or shark size. The AUC for the top selected model was actually a bit higher than the AUC values for the two models in the original manuscript (although still not amazing). The new methodology is described in lines 410 – 473. Results are presented in Table 4 (AIC model selection), Figure 3 (beta coefficients of top model), Figure 4 (predicted effects of covariates in the top model), and in the text on lines 656-663, lines 782-789.

We altered the discussion of AVS as necessary to fit the results of the revised analysis, although the required changes were not extreme as the main effects of soak time and water temperature were also important in the new analysis.

When estimating bycatch survival using our Monte Carlo simulations, the biggest change is that instead of estimating bycatch survival for two regions (WNA and GOM), we estimated bycatch survival individually for the 6 regions with > 100 mako shark observations (lines 548-552). The results largely correspond with our previous findings (greater survival in northern regions compared to the GOM), but with more spatial detail. See Fig. 6 and lines 919-924.

We feel the revised analyses satisfies reviewer concerns while providing more detailed information on survival through the inclusion of data from a larger portion of the Atlantic than the original analysis. Still, the major findings of the original manuscript remain – PRS was greater than AVS, and survival varied regionally. Thus, our overall conclusions are the same.

To address concerns voiced to some degree by all reviewers, we added information on target species and more detail on differences in fishing and environmental conditions between regions. This includes a new Table 2, which shows the proportion of observed sets in each region targeting different species, the hook depth ranges associated with each target species, and the proportion of observed mako shark captures associated with targets in each region. A new Figure 2 shows distributions of covariates of interest between the 6 major observed regions in the form of a series of box plots. Also see lines 597-605, and lines 627-641 in the results.

In our previous manuscript we included distance between gangions in the analysis of AVS. We removed that from consideration in the revised analysis. There was no evidence it had any effect and there was little biological rationale for including it.

The abstract was revised as required to correspond to other revisions.

Our responses to individual comments are below

Reviewer #1: Review: Bycatch survival of shortfin mako sharks (Isurus oxyrinchus) in the U.S. Atlantic pelagic longline fishery

This paper presents a comprehensive assessment of AVS and PRS for shortfin mako shark from US fleets in the North Atlantic. It is well-written and comprehensive, and I particularly liked the approach of using MC simulation to produce overall bycatch mortality rates. I have no concerns about research ethics. There are two changes that I think are required before the paper is suitable for publication: (1) further detail and potential changes to the analyses in relation to hook depth information, and (2) consideration of bycatch rates predicted from only the significant predictors in the GAMM. My detailed comments are provided in the attached file.

One idea that the authors could add to the Introduction is that the landings ban for mako in the North Atlantic was also intended to change fishing behaviour. The reasoning was that mako captures would cause losses (time, gear, etc.) and no monetary gain, providing an incentive to avoid interactions in the first place.

We revised the sentence on lines 121 – 123 to include a reference to changing fisher behavior. Revised sentence reads: “Retention bans aim to reduce fishing mortality to allow for population recovery by removing direct mortality associated with retaining individuals, and potentially by altering fisher behavior to avoid capturing the species.”

When quantifying the differences in fishing characteristics between the GOM and WNA, I was particularly interested in the information on hook depth. Where does this information come from? The discussion implies that hook depth information comes from POP observers, so is it possible that the relatively small differences (10s of m) reflect different estimation ability of observers in the two regions? Line 245 suggests that at least one longline set had hook depths of 2m (the lower boundary of the range), which seems improbable given the catenary curve of a mainline between floats combined with gangion lengths. Table 2 lists ‘approximate’ hook depths, yet values are presented as known elsewhere in the MS, so does this mean there are two sources? Throughout the MS, please identify and discuss how hook depth was determined and (more importantly) quantify underlying variability if the values were estimated. This would help the reader to assess if the weak relationships with hook depth from the regression analyses are meaningful, particularly given that they take different shapes among regions.

Yes, the hook depths are estimated by the observers and represent the best estimate of maximum hook depth. We better define what hook depth means in this study and describe how the depth is estimated on lines 384 – 386.

We address that a single estimate is unlikely to be fully representative of every hook in a set and that hook depth likely varies for the reasons the reviewer listed, and how that may have reduced our ability to detect a significant effect of depth on AVS in the discussion on lines 1230 -1234.

From my understanding of the MC simulation, the predicted relationships for all variables were sampled, even those that were non-significant and/or had very little evidence of a relationship. I am concerned that this approach exaggerates the differences in bycatch survival among the two regions. I would like to see an analysis using only the significant predictors from each region in the MC simulation and present the analysis now in the MS as a sensitivity analysis. If the authors have supporting literature demonstrating that their current approach is more statistically rigorous, this should be added as justification.

In the revised analysis, all covariates used in the AVS model in the MC simulation were meaningful (significant).

Specific Comments:

Line 66: Potential addition: ‘….. and that overfishing is occurring with high probability [8]’.

Done.

Line 71: perhaps ‘fully implemented’ is better terminology than ‘adopted’, given that there is always a time lag between when ICCAT recommendations are adopted and when they are fully in force in various jurisdictions.

Revised as suggested.

Line 145-150: It would be worth mentioning why the modeling was split into two components, first a GLMM and then region-specific GAMMs. Presumably because this allowed for different relationships with predictor variables among regions?

In the revised analysis we rely entirely on GLMMs.

Line 145-150: Were any interactions tested in the suite of models? After reading the results, I was interested if there was any evidence for an interaction between SST and soak time for WNA. This would also bolster the information in the discussion (lines 402-416).

In our revised AVS model we did consider interactions in our candidate models, mentioned on lines 418-420, and all candidate models are presented in Table 4.

Line 160: Is 0.7 a common threshold for collinearity? Please provide a supporting reference.

Yes, it is. We included the reference:

Dormann CF, Elith J, Bacher S, Buchmann C, Carl G, Carré G, García Marquéz JR, Gruber B, Lafourcade B, Leitão PJ, et al. (2013) Collinearity: a review of methods to deal with it and a simulation study evaluating their performance. Ecography 36(1): 27–46

Line 187: Does the US fleet use wire leaders with weights or monofilament?

Monofilament, at least in the Atlantic. We revised to “monofilament leader” here (line 505).

Line 217: Section on Bycatch survival. It would be helpful to a reader to have a specific statement on why the simulation was done. This would highlight how your analysis better captures uncertainty in both estimated rates when generating the overall value, as compared to other analyses that merely multiply the individual means.

Good idea. We added text explaining this on lines 569 – 593.

Line 251: Were records considered incomplete if a single covariate was missing?

For the analysis examining the influence of covariates on AVS we only included records where all covariates were available. We clarify this on lines 415 – 418.

Line 265: ‘…. holding other variables constant at their predicted mean, …..’ Similar comment for Line 302.

This specific line does not exist in the revised manuscript, but we used the suggested language consistently in the revised results on lines 782 – 789.

Line 353: consider replacing ‘=’ with ‘of’.

Revised as suggested.

Lines 355-357 Re: bycatch survival. It would help the reader to have a simple statement that these are overall estimates (AVS*PRS).

Revised to “Monte Carlo simulations combining our estimates of AVS and PRS suggested…” (line 919).

Line 418: Figure 1 doesn’t show the depth ranges that sharks were hooked at in the NWA or GOM. Maybe reference Figure 2?

Good catch. We checked to make sure we reference figures appropriately in the revised manuscript

Line 417-418: For an animal that regularly dives from the surface to > 400 m in a day (Figure 5), I have a hard time with the idea that the difference in mean hook depth between the two regions is biologically meaningful. Typically shallow-set longline is differentiated from deep-set longline by 100s not 10s of meters, with the associated expectation of differences in AVS and PRS.

In the revised analysis depth was not found to be informative. Regional variation in hook depth is presented in figure 2 of the revised manuscript.

Line 536-537: Please reword to ‘…. as possible are associated with very high survival probabilities.’ Without assessing covariate effects, it is not possible to say that these actions maximize survival probabilities.

Revised as suggested – line 1415.

Figure 2: There seem to be no values on the x axes on each panel?

This figure does not exist in the revised manuscript. But we were sure to include appropriate values for all axes.

Supplementary File S2. Has the data been cleaned to only represent diving behaviour? Or are there still the portions of the tracks that represent times when the tag was at constant depth and/or drifting at the surface? Please clarify.

Revised to indicate it is the full track.

Reviewer #2: This study presents the results of an important data collection on the survival of mako sharks caught by commercial pelagic longline fisheries (US) in the Northwest Atlantic. In this study, the authors assess the survival (i.e. at-vessel survival AVS and post-release survival PRS) of the mako shark as a bycatch specie in the LL fisheries and identify the effects of environmental, biological and fishing practices variables on at-vessel survival. Given their results, the authors discuss the retention ban as a bycatch mitigation measure to the restoration of the abundance level of overfished population.

The manuscript is interesting because it provides important and much-needed data on total mortality (at-vessel and post-release survival) in a species that is highly vulnerable to pelagic longline fisheries, providing a more complete analysis of the impact of this fishery on mako sharks. It also tackles the issue of pelagic sharks’ fishing mortality in a new way through a study focusing on a comparison between two adjacent fishing regions of the north-western Atlantic.

However, the manuscript lacks scientific rigor at several levels:

The manuscript requires improvement in its use of standard English and scientific writing. Some sentences are not complete. (See my detailed comments below for specific examples).

General connections and shortcuts are made between different notions of fisheries, without the underlying logical links being clearly explained (e.g. stock assessment, risk assessment, overall mortality, retention ban...).

It is unclear what the reviewer is referring to here. As such, it is hard to make any edits. We did remove the term “risk assessment” from line 955.

The manuscript lacks consistency in the use of the terms “mortality” and “survival”. I think the authors should choose one or the other, but not both as it becomes confusing to follow the logical reasoning.

We understand how this could be confusing. There is variability in what is reported (survival or mortality) in the literature that is cited. To help ameliorate this issue, starting on line 129, after introducing and defining AVM and PRM, we define AVS and PRS, then refer to survival for the rest of the manuscript. Note that when referring to other literature, this sometimes required us to calculate survival when a cited study only presented mortality. It is a

---

## [Editor Report · Decision Letter 1]

10 Aug 2025

Bycatch survival of shortfin mako sharks (*Isurus oxyrinchus* ) in the U.S. Atlantic pelagic longline fishery

PONE-D-24-54309R1

Dear Dr. Byrne,

We’re pleased to inform you that your manuscript has been judged scientifically suitable for publication and will be formally accepted for publication once it meets all outstanding technical requirements.

Kind regards,

Rui Coelho, PhD

Academic Editor

PLOS ONE

Additional Editor Comments (optional):

I believe that the authors have addressed the issues identified by the reviewers, and as such I recommend that the paper can be accepted for publication.
---

## [Editor Report · Acceptance letter]

PONE-D-24-54309R1

PLOS ONE

Dear Dr. Byrne,

I'm pleased to inform you that your manuscript has been deemed suitable for publication in PLOS ONE. Congratulations! Your manuscript is now being handed over to our production team.

Kind regards,

on behalf of

Dr. Rui Coelho

Academic Editor

PLOS ONE